# GENERATIVE PSEUDO-INVERSE MEMORY

**Kha Pham** [1], **Hung Le** [1], **Man Ngo** [2], **Truyen Tran** [1], **Bao Ho** [3] **and Svetha Venkatesh** [1]

[1] Applied Artificial Intelligence Institute, Deakin University

[2] Faculty of Mathematics and Computer Science, VNUHCM-University of Science

[3] Vietnam Institute for Advanced Study in Mathematics

[1] `{phti, thai.le, truyen.tran, svetha.venkatesh}@deakin.edu.au,`

[2] `nmman@hcmus.edu.vn,`

[3] `bao@viasm.edu.vn`

## ABSTRACT

We propose Generative Pseudo-Inverse Memory (GPM), a class of deep generative memory models that are fast to write in and read out. Memory operations are recast as seeking robust solutions of linear systems, which naturally lead to the use of matrix pseudo-inverses. The pseudo-inverses are iteratively approximated, with practical computation complexity of almost $O(1)$. We prove theoretically and verify empirically that our model can retrieve exactly what have been written to the memory under mild conditions. A key capability of GPM is iterative reading, during which the attractor dynamics towards fixed points are enabled, allowing the model to iteratively improve sample quality in denoising and generating. More impressively, GPM can store a large amount of data while maintaining key abilities of accurate retrieving of stored patterns, denoising of corrupted data and generating novel samples. Empirically we demonstrate the efficiency and versatility of GPM on a comprehensive suite of experiments involving binarized MNIST, binarized Omniglot, FashionMNIST, CIFAR10 & CIFAR100 and CelebA.

## 1 INTRODUCTION

Memory is central to intelligence by facilitating information compression, reconstruction, manipulation, and generation. The processing speed and storage capacity of the working memory are known to correlate with reasoning capacity (Jensen and Munro, 1979; Kyllonen and Christal, 1990). Recent work in machine learning has explored slot-based external memory to augment neural networks (Graves et al., 2016; Le et al., 2019) in which memory reads and writes proceed sequentially through attention mechanism, making training and pattern retrieval difficult for very long sequences. Global self-attention techniques (Ramsauer et al., 2021) create short-cuts in information paths, hence are easier to train, but require large memory and computation for long sequences.

Different from slot-based memories, Kanerva Machines (Wu et al., 2018a;b) are a class of generative memory inspired by Kanerva's sparse distributed memory (Kanerva, 1984). Here memory update/retrieval and addressing mechanism are treated as Bayesian inference where posteriors are updated when a new data episode arrives. This iterative and dynamic inference handles noisy inputs better due to the convergence of the attractor dynamics to fixed points (Wu et al., 2018b). However, these models suffer from slow processing speed, which originates from the sequential writing mechanism. This slowness prevents the model to adapt to large batch of data, thus makes it difficult to conduct experiments on memory storage capacity. On the other side, one cannot ensure theoretically how well those models retrieve information. This creates an obstacle for theoretical research on the efficiency of memory models.

To overcome these challenges we propose a new model called Generative Pseudo-Inverse Memory (GPM), which is illustrated in Fig. 1. GPM reformulates the Bayesian updates of memory and address as finding least-square solutions to linear systems. Among these solutions, the smallest norm is found through applying the Moore-Penrose *pseudo-inverse* of matrices (Ben-Israel and Greville, 2001). This helps us achieve rapid and accurate memory read/write. Moreover, our model is able to not only store and retrieve information *perfectly under ideal conditions* but also generate new samples based

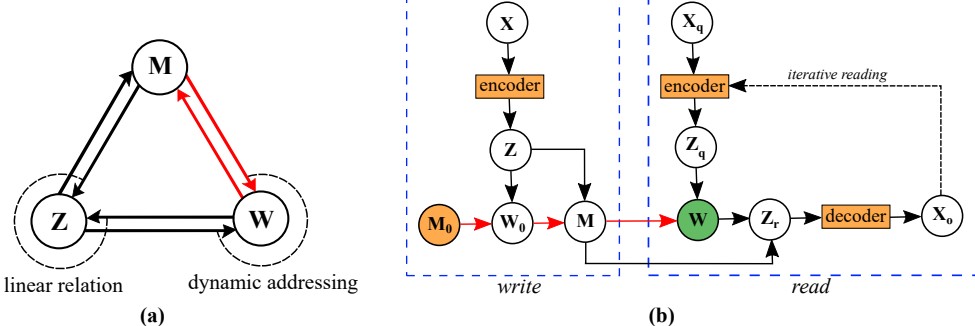

Figure 1: **(a)** *Folded writing-reading mechanism of GPM.* Red arrows indicates operations of computing the pseudo-inverses. The memory $\mathbf{M}$, the weight $\mathbf{W}$ and the data $\mathbf{Z}$ interact in a dynamic way, where $\mathbf{W}$ is dynamically computed based on $\mathbf{Z}$ and $\mathbf{M}$ and (assume that) $\mathbf{Z}$ depends linearly on $\mathbf{W}$ and $\mathbf{M}$. **(b)** *Unfolded writing-reading mechanism of GPM.* Orange boxes and circles contain trainable parameters of GPM. The green circle consists of variational part for the generative scheme. Red arrows are operations of computing the pseudo-inverses. During writing, the episode $\mathbf{X}$ is encoded as $\mathbf{Z}$. Using $\mathbf{Z}$ and prior memory $\mathbf{M}_0$, we compute the writing weight $\mathbf{W}_0$ and then use $\mathbf{W}_0$ to write $\mathbf{Z}$ into the memory. The result of writing phase is the posterior memory $\mathbf{M}$. In reading, the query $\mathbf{X}_q$ is encoded as $\mathbf{Z}_q$. Note that the encoder in the reading phase needs not be identical to the encoder in writing phase. The reading weight $\mathbf{W}$ is computed based on $\mathbf{Z}_q$ and the posterior $\mathbf{M}$. The memory read-out $\mathbf{Z}_r$ is simply the matrix product between $\mathbf{W}$ and $\mathbf{M}$. Finally, $\mathbf{Z}_r$ goes through the decoder to produce the output $\mathbf{X}_o$ of the reading phase. If iterative reading is included, $\mathbf{X}_o$ will become the next query and be fed back to the encoder for next reading step. In training, we use the same encoder for writing and reading, the query $\mathbf{X}_q$ is identical with $\mathbf{X}$ and reading is non-iterative.

on written data. This coincides with recent inspiring results from functional neuroimaging (Addis et al., 2007; Okuda et al., 2003; Schacter et al., 2012), which showed that memory not only helps us memorize but also supports imagining. We derive an energy function for GPM and show that its updates seek to locate local minima of the energy. This dynamic enables GPM to perform iterative reading to improve sample quality during denoising and generating. Finally, GPM can store a large amount of data (up to a dataset) while still maintaining key capabilities. We validate these theoretical insights through a comprehensive suite of experiments on binarized MNIST (LeCun et al., 2010) , binarized Omniglot (Burda et al., 2016), FashionMNIST (Xiao et al., 2017), CIFAR10 & CIFAR100 (Krizhevsky, 2009) and CelebA (Liu et al., 2015), demonstrating superior results.

## 2 MODEL

Consider a data episode of length $T$ and let $\mathbf{X}$ denote the episode data matrix of $T$ rows, where the $i$-th row of $\mathbf{X}$ is the $i$-th item of the episode. Throughout this paper, we refer to the episode as its associated episode matrix. We aim to compress the episode into a fixed-size memory for later reconstruction and generation processes. We maintain a memory matrix $\mathbf{M}$, which is a deterministic real matrix of size $K \times C$ with trainable initialization $\mathbf{M}_0$. For convention, $\mathbf{M}_0$ may be considered as the common prior memory containing information of the whole dataset while $\mathbf{M}$ is the posterior memory dependent on the episode.

*Remark.* In this paper, we reuse classical terms "prior memory" and "posterior memory", which are frequently used in prior works where the memory is modeled as a distribution. In this paper, however, we only consider deterministic memories with an exception when we introduce the optimization problem corresponding to the Bayesian inference (Eq. (3)). Hence, "prior memory" and "posterior memory" should be interpreted as the memory *before* and *after* data arrive, respectively.

In order to write to and read from the memory, we additionally define the writing weight matrix $\mathbf{W}_0$ and reading weight matrix[1] $\mathbf{W}$, both of size $T \times K$. We treat the reading weight $\mathbf{W}$ as a random variable, hence inducing the generative capability of the model. For simplicity, the prior $p\left(\operatorname{vec}(\mathbf{W})\right)$ of $\mathbf{W}$ is the standard Gaussian $\mathcal{N}\left(\mathbf{0}, \mathbf{I}_{T \times K}\right)$ while the posterior $q\left(\operatorname{vec}(\mathbf{W})\right)$ is $\mathcal{N}\left(\overline{\mathbf{W}}, \sigma_{\mathbf{W}}^2 \mathbf{I}_{T \times K}\right)$, where the mean $\overline{\mathbf{W}}$ is adapted to each episode and $\sigma_{\mathbf{W}}$ is a learnable parameter. Read-outs from the memory are matrix products between the reading weight and the posterior memory.

---

[1]We note that weights here are dynamic and unrelated to neural network parameters. The term "dynamic" means the writing/reading weights of an input not only depend on its own but also the episode it belongs to.

Let $e_\theta(.)$ be the (neural) encoder and $d_\theta(.)$ be the decoder parameterized by $\theta$. In the following we drop the explicit subscript $\theta$ for clarity. We further let $\mathbf{Z}$ denote the $T \times C$ encoding matrix of the episode $\mathbf{X}$, i.e. $\mathbf{Z} = e(\mathbf{X})$ where $e(.)$ is applied to each row of $\mathbf{X}$. During training, the model can only observe a noisy version of $\mathbf{Z}$, which we denote by $\mathbf{Z}_\xi$. Specifically, $\mathbf{Z}_\xi = \mathbf{Z} + \xi$, where $\xi \sim \mathcal{N}(\mathbf{0}, \sigma_\xi^2 \mathbf{I})$ is the observation noise. We assume that the observed data $\mathbf{Z}_\xi$ depends linearly on the weights and the memories: $\mathbf{Z}_\xi = \mathbf{W}_0 \mathbf{M}_0$ and $\mathbf{Z}_\xi = \mathbf{W}\mathbf{M}$, which we refer as *linear assumptions over* $\mathbf{Z}_\xi$.

## 2.1 Objective function

Given the memory $\mathbf{M}$, we aim to maximize the conditional log-likelihood $\ln p(\mathbf{X}|\mathbf{M})$. It is difficult to maximize $\ln p(\mathbf{X}|\mathbf{M})$ directly, so we instead maximize its evidence lower bound $\mathcal{L}$, where

$$\mathcal{L} = \mathbb{E}_{q(\mathbf{W})} \ln p(\mathbf{X}|\mathbf{W}, \mathbf{M}) - D_{\mathrm{KL}}(q(\mathbf{W})||p(\mathbf{W})). \tag{1}$$

The first term in $\mathcal{L}$ is usually called the negative reconstruction loss, while the second term is the Kullback-Leibler divergence between the posterior $q(\mathbf{W})$ and the prior $p(\mathbf{W})$ of the reading weight. Details of derivation will be given in Appendix I.

In our model, we use the encoding matrix $\mathbf{Z}$ as the data to be written to the memory instead of the original episode $\mathbf{X}$. Therefore, we would like $\mathbf{Z}$ to be a "good" representation of $\mathbf{X}$. We quantify this goodness by the usual auto-encoder loss $\mathcal{L}_{\mathrm{AE}} = \mathbb{E}_{\mathbf{X} \sim \mathrm{data}} \ln p\left(d\left(e(\mathbf{X})\right)\right)$. Thus during training, we will maximize the objective function:

$$\mathcal{O} = \mathcal{L} + \mathcal{L}_{\mathrm{AE}}. \tag{2}$$

## 2.2 Generative Pseudo-Inverse Memory

In our model, the posterior memory, the writing weight and the reading weight are all dynamic (i.e. depending on the data episode). Hence it is necessary to understand the motivations and computation methods as introduced in the following.

**Intuition**

Only in this part, we will consider probabilistic memories to have unified settings with previous Bayesian-based works such as the Kanerva Machines (Wu et al., 2018a;b; Marblestone et al., 2020). When the observed data $\mathbf{Z}_\xi$ arrive, the Kanerva Machines apply the Bayesian inference process to obtain the posterior memory from a given (trainable) prior distribution. It is well-known that the posterior mean is the Bayes estimator with respect to the squared error risk (Jaynes, 2003). Formally, given the observed data $\mathbf{Z}_\xi$ and the writing weight $\mathbf{W}_0$, the posterior memory mean $\mathbf{M}$ is the solution of the optimization problem

$$\min_{\mathbf{M}} \mathbb{E}\left( \|\mathbf{M} - \mathbf{M}_0\|_F^2 \Big| \mathbf{Z}_\xi, \mathbf{W}_0 \right), \tag{3}$$

where $\|.\|_F$ is the Frobenius norm. With the linear assumption over $\mathbf{Z}_\xi$ and an additional linear assumption over $\mathbf{M}$, Eq. (3) has a unique and explicit solution (see Appendix H).

The solution $\mathbf{M}$ of Eq. (3) tends to preserve information from $\mathbf{M}_0$. However, in practice, what appears in the prior memory might be blurry information. Forcing the short-term memory $\mathbf{M}$ to be close to $\mathbf{M}_0$ may prevent $\mathbf{M}$ from storing sufficient information, leading to incorrect read-out.

**Memory inference**

From now on, we will only consider deterministic memories. We propose a weighted version of the optimization problem (3) to help improve read-out accuracy. Our proposed optimization problem is

$$\min_{\mathbf{M}} \|\mathbf{W}_0(\mathbf{M} - \mathbf{M}_0)\|_F^2. \tag{4}$$

Intuitively, if $\mathbf{M}$ is a solution of Eq. (4), then $\mathbf{W}_0\mathbf{M}$ is close to $\mathbf{W}_0\mathbf{M}_0$, which is equal to $\mathbf{Z}_\xi$ due to the linear assumption over $\mathbf{Z}_\xi$. In other words, the read-out after writing (i.e. $\mathbf{W}_0\mathbf{M}$) is similar to the data written to the memory (i.e. $\mathbf{Z}_\xi$). Substitute $\mathbf{W}_0\mathbf{M}_0 = \mathbf{Z}_\xi$, Eq. (4) becomes

$$\min_{\mathbf{M}} \|\mathbf{W}_0\mathbf{M} - \mathbf{Z}_\xi\|_F^2. \tag{5}$$

Any solution of Eq. (5) is called the *least-square solution* of the linear system $\mathbf{W}_0\mathbf{M} = \mathbf{Z}_\xi$. However, Eq. (5) may admit more than one least-square solution, thus we are interested in the least-square solution that has the smallest norm, i.e. the *minimum-norm least-square solution*. The minimum-norm condition may help to regularize in the sense that the posterior memory should contain as little redundant information as possible. It can be proved that $\mathbf{M} = \mathbf{W}_0^+\mathbf{Z}_\xi$ is the minimum-norm least-square solution of the system $\mathbf{W}_0\mathbf{M} = \mathbf{Z}_\xi$ (Ben-Israel and Greville, 2001), where $\mathbf{W}_0^+$ is the *pseudo-inverse* of $\mathbf{W}_0$. The pseudo-inverse of matrix is applicable to classical associative memory which involves finding a linear mapping from input to output (Stiles and Denq, 1985; Yen and Michel, 1991). Readers are referred to Appendix D for more mathematical properties of the pseudo-inverse.

**Computing writing and reading weights (dynamic addressing)**

The discussion so far is about seeking the posterior memory $\mathbf{M}$ given the data $\mathbf{Z}$ and the writing weight $\mathbf{W}_0$. In our settings, all weights are dynamic and they depend on either the prior memory and data (during writing) or the posterior memory and the query (during reading). Formally written, we need to determine the writing weight $\mathbf{W}_0$ (given prior memory $\mathbf{M}_0$ and noisy data $\mathbf{Z}_\xi$) and reading weight $\mathbf{W}$ (given posterior memory $\mathbf{M}$ and query $\mathbf{Z}$)[2]. Ideally, $\mathbf{W}_0$ and $\mathbf{W}$ should satisfy $\mathbf{W}_0\mathbf{M}_0 = \mathbf{Z}_\xi$ and $\mathbf{W}\mathbf{M} = \mathbf{Z}_\xi \approx \mathbf{Z}$, according to the linear assumptions over $\mathbf{Z}_\xi$. The minimum-norm least-square solutions of those linear systems are $\mathbf{W}_0 = \mathbf{Z}_\xi\mathbf{M}_0^+$ and $\mathbf{W} = \mathbf{Z}\mathbf{M}^+$, where $\mathbf{M}_0^+$ and $\mathbf{M}^+$ are pseudo-inverses of $\mathbf{M}_0$ and $\mathbf{M}$, respectively. While least-square condition is obvious since we would like to find solutions $\mathbf{W}_0$ and $\mathbf{W}$ that best fit the systems, explanation for the minimum-norm condition will be given in Appendix J.

**Computing pseudo-inverses**

We do not always have an explicit formula for the pseudo-inverse, and even in such cases, the formula may consist of inverse matrices which are expensive to compute. Ben-Israel and Cohen (1966) provide us with an iterative way to approximate the pseudo-inverse, as stated in the following theorem:

**Theorem 2.1.** (Ben-Cohen algorithm) *Given a real matrix $\mathbf{D}$ and an initial matrix $\mathbf{D}_0$ of the same size with the transpose $\mathbf{D}^\top$. The sequence $\{\mathbf{D}_i\}_{i \geq 0}$ defined recursively as $\mathbf{D}_{i+1} = 2\mathbf{D}_i - \mathbf{D}_i\mathbf{D}\mathbf{D}_i$ will converge to the pseudo-inverse of $\mathbf{D}$. With appropriate $\mathbf{D}_0$, the sequence will converge quadratically.*

Ben-Cohen algorithm allows us to compute the matrix pseudo-inverse efficiently with nearly $O(1)$ time complexity in practice (see Section 3.5 for experimental results). For simplicity, we set the initial term $\mathbf{D}_0 = \alpha\mathbf{D}^\top$, where $\alpha$ is a hyper-parameter dependent on the dataset. Deeper analysis of the initial condition for Ben-Cohen algorithm will be given in Appendix G.

**Algorithm**   We are now ready to present the Generative Pseudo-Inverse Memory (GPM) in full. Assume that the observation noise $\xi$ is sampled from $\mathcal{N}(\mathbf{0}, \sigma_\xi^2\mathbf{I})$. Algorithm 1 illustrates a single training step of GPM. All pseudo-inverses are approximated by the Ben-Cohen algorithm.

### 2.2.1   REMARK: RELATION WITH EXISTING ALGORITHMS

Several properties of GPM are worth highlighting. GPM can be viewed as 2-step EM algorithm (Dempster et al., 1977). Details are given in Appendix J. Moreover, in noiseless setting, i.e., $\xi \to 0$, GPM and Dynamic Kanerva Machine (DKM) (Wu et al., 2018b) coincide. In DKM, solving $\overline{\mathbf{W}}$ is related to minimizing $D_{\mathrm{KL}}\left(q(\mathbf{W})||p(\mathbf{W}|\mathbf{X}, \mathbf{M})\right)$, which is equivalent to the optimization problem

$$\min_{\overline{\mathbf{W}}} \left( \|\mathbf{Z} - \overline{\mathbf{W}}\mathbf{M}\|_F^2 + \sigma_\xi^2\|\overline{\mathbf{W}}\|_F^2 \right). \tag{6}$$

This problem has a solution $\overline{\mathbf{W}}_{\sigma_\xi} = \mathbf{Z}\mathbf{M}^\top(\mathbf{M}\mathbf{M}^\top + \sigma_\xi^2\mathbf{I})^{-1}$. According to Theorem D.5, $\overline{\mathbf{W}}_{\sigma_\xi}$ converges to $\overline{\mathbf{W}}^* = \mathbf{Z}\mathbf{M}^+$ as $\sigma_\xi \to 0$. Note that $\overline{\mathbf{W}}^*$ is actually the reading weight of GPM.

### 2.3   ERROR BOUND

Using pseudo-inverses in Algorithm 1, we establish an error bound for the memory read-out, as stated in the following theorem:

---

[2]In general case (e.g. when the query is a noisy version of $\mathbf{Z}$), the same dynamic addressing holds.

---

**Algorithm 1** Single training step of Generative Pseudo-Inverse Memory

---

Sample an episode $\mathbf{X}$ of length $T$.

**Writing**

1. Compute episode embedding $\mathbf{Z} = e(\mathbf{X})$.
2. Randomize noise $\xi$ from $\mathcal{N}(\mathbf{0}, \sigma_\xi^2 \mathbf{I})$.        // simulate observed data
3. Compute weight $\mathbf{W}_0 = \mathbf{Z}_\xi \mathbf{M}_0^+$.        // dynamic addressing
4. Compute posterior memory $\mathbf{M} = \mathbf{W}_0^+ \mathbf{Z}_\xi$.        // complete writing phase

**Reading**

1. Compute episode embedding $\mathbf{Z} = e(\mathbf{X})$.    // similar role with $\mathbf{Z}_q$ in Figure 1b
2. Compute weight mean $\overline{\mathbf{W}} = \mathbf{Z}\mathbf{M}^+$.        // dynamic addressing
3. Sample $\mathbf{W} \sim \mathcal{N}(\overline{\mathbf{W}}, \sigma_\mathbf{W}^2 \mathbf{I})$.    // variational schema for generating samples
4. Compute read-out $\mathbf{Z}_{\text{read-out}} = \mathbf{W}\mathbf{M}$.    // similar role with $\mathbf{Z}_r$ in Figure 1b
5. Compute reconstruction $\mathbf{X} = d(\mathbf{Z}_{\text{read-out}})$.  // complete reading phase, no iterative reading

**Updating model parameters**

1. Compute the objective $\mathcal{O} = \mathcal{L} + \mathcal{L}_{AE}$ using obtained terms from previous steps.
2. Update parameters via gradient ascent to maximize $\mathcal{O}$.

---

**Theorem 2.2.** *Let $\|.\|_2$ denote the spectral norm. Suppose there exist $\alpha, \beta \in [0, 1)$ such that*

$$\|\mathbf{I} - \mathbf{Z}_\xi \mathbf{Z}_\xi^\top\|_2 \leq \alpha \text{ and } \|\mathbf{I} - \mathbf{Z}_\xi \mathbf{Z}_\xi^+\|_2 \leq \beta.$$

*Then $\overline{\mathbf{W}}\mathbf{M} = \mathbf{Z}(\mathbf{I} + \mathbf{E})$, where $\mathbf{E}$ is the error matrix satisfying*

$$\|\mathbf{E}\|_2 \leq \beta + \|\mathbf{I} - \mathbf{W}_0 \mathbf{W}_0^+\|_2 \sqrt{\tfrac{1+\alpha}{1-\alpha}}.$$

**Corollary 2.3.** *Assume that rank $\mathbf{W}_0 =$ rank $\mathbf{Z}_\xi = T$, i.e. rows of $\mathbf{W}_0$ and $\mathbf{Z}$ are linearly independent. Then $\overline{\mathbf{W}}\mathbf{M} = \mathbf{Z}$.*

Proofs are given in Appendix E. Theorem 2.2 and Corollary 2.3 establish that the error of memory retrieval can be bounded by other errors only dependent on the data $\mathbf{Z}$ and writing weight $\mathbf{W}_0$. When rows of $\mathbf{W}_0$ and $\mathbf{Z}_\xi$ are linearly independent, *the retrieval is perfect*. The trainable prior memory $\mathbf{M}_0$ helps the model balance approximation errors when ideal conditions are not met.

Corollary 2.3 also suggests that retrieval may be still accurate if rows of $\mathbf{W}_0$ and $\mathbf{Z}$ are sufficiently linearly independent. A well-known property on high dimensional space is that two arbitrary vectors are likely to be orthogonal (Gorban and Tyukin, 2018). Thus we can expect that GPM will still work fairly well when $T \gg \max\{K, C\}$, i.e. the episode length is much greater than the number of memory slots and size of embedding vectors. Empirical evidences will be given in Section 3.4.

## 2.4 ATTRACTOR DYNAMICS

We demonstrate that GPM inherits properties of energy-based models. The key property of models induced from dynamical systems is iterative reading, i.e. to iteratively reproduce the correct pattern from a partially broken query. With fixed memory $\mathbf{M}$, the energy function of GPM can be defined as a function of two variables $\mathbf{x}$ and $\mathbf{w}$:

$$\mathcal{E}(\mathbf{x}, \mathbf{w}) := -\ln p(\mathbf{x}|\mathbf{w}, \mathbf{M}) = -p(\mathbf{x}|\mathbf{M}) + \frac{\|e(\mathbf{x}) - \overline{\mathbf{w}}\mathbf{M}\|_F^2}{2\sigma_\xi^2} + \mathbf{r}. \tag{7}$$

Here we replaced $\mathbf{z}$ by $e(\mathbf{x})$ and note that the residual $\mathbf{r}$ is constant after training. Denote $(\mathbf{w}_i, \mathbf{x}_i)$ the pair of weight and reading output of the $i$-th iterative reading step. The output $\mathbf{x}_i$ is iteratively fed back to the model until convergence. With $\mathbf{x}_i$ as the query for $(i+1)$-th step, solving for $\mathbf{w}_{i+1}$ minimizes $\|e(\mathbf{x}) - \overline{\mathbf{w}}\mathbf{M}\|_F^2$, which indicates $\mathcal{E}(\mathbf{x}_i, \mathbf{w}_{i+1}) > \mathcal{E}(\mathbf{x}_i, \mathbf{w}_i)$. With $\mathbf{w}_{i+1}$ in hand, the way we train the model encourages $\mathbf{x}_{i+1}$ to maximize the likelihood $p(\mathbf{x}|\mathbf{w}_{i+1}, \mathbf{M})$. If it is the case, $\mathcal{E}(\mathbf{x}_{i+1}, \mathbf{w}_{i+1}) > \mathcal{E}(\mathbf{x}_i, \mathbf{w}_{i+1})$ and therefore $\mathcal{E}(\mathbf{x}_{i+1}, \mathbf{w}_{i+1}) > \mathcal{E}(\mathbf{x}_i, \mathbf{w}_i)$. In ideal cases, $\{\mathbf{x}_i\}_{i \geq 0}$ will converge to a local minimum associated with a stored pattern in the energy landscape.

| Method | Binarized MNIST 28 × 28 (nats/image) | Binarized Omniglot 28 × 28 (nats/image) | CIFAR10 32 × 32 (bits/dim) |
|---|---|---|---|
| VAE (Kingma and Welling, 2014) | 87.86 | 104.75 | 6.3 |
| IWAE (Burda et al., 2016) | 85.32 | 103.38 | - |
| **Improved decoders** | | | |
| PixelVAE++ (Sadeghi et al., 2019) | 78.00 | - | **2.90** |
| MAE (Ma et al., 2019) | 77.98 | 89.09 | 2.95 |
| DRAW (Gregor et al., 2015) | 87.4 | 96.5 | 3.58 |
| MatNet (Bachman, 2016) | 78.5 | 89.5 | 3.24 |
| **Richer priors** | | | |
| Ordered ACN (Graves et al., 2018) | 73.9 | - | 3.07 |
| VLAE (Chen et al., 2017) | 78.53 | 102.11 | 2.95 |
| VampPrior (Tomczak and Welling, 2018) | 78.45 | 89.76 | - |
| **Memory models** | | | |
| VMA (Bornschein et al., 2017) | - | 103.6 | - |
| KM (Wu et al., 2018a) | - | 68.3 | 4.37* |
| DNC (Graves et al., 2016) | - | 100 | - |
| DKM (Wu et al., 2018b) | 75.3 | 77.2 | 4.79 |
| Kanerva++ (Ramapuram et al., 2021) | 41.58 | 66.24 | 3.28 |
| GPM (ours) | **31.48** | **25.68** | 4.03 |

Table 1: *Negative evidence lower bound of test likelihood (lower is better).* * Training ELBO, estimated from Figure 12 in Wu et al. (2018a).

In practice, the system might not converge to a stored pattern but a spurious state, which is a local minimum different from stored patterns. Even with simple neural networks in low dimensional space, spurious local minima also exist (Safran and Shamir, 2018). This phenomenon is not expected when retrieving, but it may help generate new images as demonstrated in Section 3.3.

# 3 EXPERIMENTS

We examine GPM on common generative memory benchmarking datasets including binarized MNIST (LeCun et al., 2010), binarized Omniglot (Burda et al., 2016), FashionMNIST (Xiao et al., 2017), CIFAR10, CIFAR100 (Krizhevsky, 2009) and CelebA (Liu et al., 2015). To prove the effectiveness of our memory model, we only use simple encoders and decoders. The details of image preprocessing, network architecture and training procedure are given in Appendix C. Codes are available at https://github.com/phamtienkha/generative-pseudoinverse-memory.

We compare our model to recent generative memory models such as Kanerva Machine (KM), Dynamic Kanerva Machine (DKM) and Kanerva++. In GPM, most of the parameters come from the encoder and the decoder since the trainable prior memory $\mathbf{M}_0$ and the weight variance $\sigma_{\mathbf{W}}^2$ only account for only $KC + 1$ parameters. Hence, the number of parameters of GPM is comparable to other baselines with similar architectures of encoder and decoder.

## 3.1 RECONSTRUCTION

In this task, we validate the memory model's ability to reconstruct image data. For each dataset, we sample 32 images to form the episode as the input to the memory models during both encoding and decoding phases. We then train all the models to optimize the objective in Eq. (2) for 10,000-15,000 iterations, depending on the dataset. Table 1 reports the negative evidence lower bound (Eq. (1)) of GPM and that of other baselines on 3 datasets, including the binarized MNIST, binarized Omniglot and CIFAR10. Results for the FashionMNIST, CIFAR100 and CelebA datasets are reported in Table 3 in Appendix A. For binarized datasets, we see a significant improvement in the ELBO: **31.48 nats/image** compared to current state-of-the-art 41.58 nats/images on MNIST dataset and **25.68 nats/image** compared to current state-of-the-art 66.24 nats/image on Omniglot dataset. On RGB datasets, GPM shows competitive result. We note that for complex image data, the role of the encoder and decoder is very important and GPM's ones are simpler than those of Kanerva++. Hence, we underperform Kanerva++ on CIFAR10 dataset. However, compared to models with similar encoder-decoder like KM and DKM, GPM performs much better, which indicates the benefit of our pseudo-inverse memory mechanisms.

To verify that our GPM supports exact memory retrieval, we measure the cosine similarity between the data written to the memory and memory read-out's of GPM in Omniglot task (see Fig. 6 of

| | salt & pepper noise (15%) | | | block noise (12 × 12) | | | rotation noise (30°) | | |
|---|---|---|---|---|---|---|---|---|---|
| | $T = 4$ | $T = 8$ | $T = 16$ | $T = 4$ | $T = 8$ | $T = 16$ | $T = 4$ | $T = 8$ | $T = 16$ |
| GPM (ours) | **40.59%** | **37.75%** | **16.89%** | **57.42%** | **40.10%** | **18.69%** | **66.09%** | **57.55%** | **39.98%** |
| DKM (Wu et al., 2018b) | 5.20% | 1.24% | 0.61% | 7.18% | 2.10% | 1.18% | 7.67% | 4.33% | 1.79% |

Table 2: *Denoising success rate on binarized Omniglot dataset.* A denoising process is successful if it can retrieve the original image with at most 1% error, which is equivalent to 7 pixels.

Appendix). As expected, GPM almost retrieves exactly what was written to the memory (average similarity $\approx 1$), which is much better than that of Dynamic Kanerva Machine (0.82).

## 3.2 DENOISING

The denoising capability of a memory model resembles human's ability to retrieve old memories with blurry clues. This process can be implemented using iterative reading, where in each step the model adds some more details to the image to reconstruct the original picture from a noisy query image. Illustration for the denoising process of GPM is given in Appendix B.

To measure the denoising capability of GPM and DKM, we run denoising experiments with different episode lengths, types of noise and levels of noise on the binarized Omniglot dataset. We use the Hamming distance to measure the error between denoised images and correct ones. In all experiments, we run iterative reading for 20 steps. We report results for the case $T = 16$ in Fig. 2, while results for other cases can be found in Fig. 8 of Appendix. Overall, GPM achieves a significantly better performance than that of DKM. Remarkably, GPM maintains a clear performance gap when the episode lengths or the level of noise vary. To better quantify the difference, we reported the denoising success rate of GPM and DKM in Table 2. Here we consider a denoising process to be successful if it can recover the original image with at most 1% error, which is equivalent to at most 7 pixels for the Omniglot dataset. The results show that GPM also outperforms DKM in all cases with a performance gap ranging from 15% to 58%.

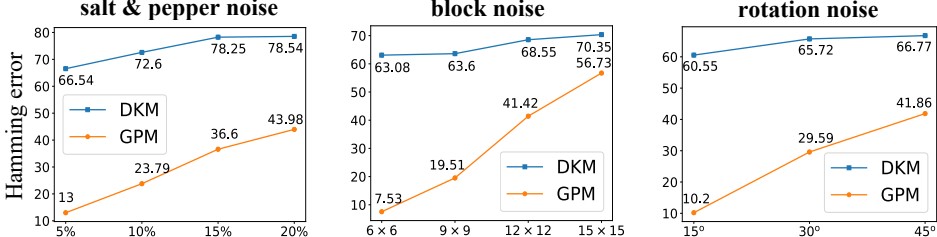

Figure 2: *Hamming error measuring the denoising capibility of GPM and DKM (lower is better) with $T = 16$ on Omniglot dataset.* We test denoising capability of both models with different types of noise (salt & pepper noise, block noise, rotation noise) and levels of noise ($5\% - 20\%$ salt & pepper noise, block noise with block size from $6 \times 6$ to $15 \times 15$, rotation noise with rotation angle from $15°$ to $45°$).

## 3.3 GENERATION

We illustrate the ability to generate new images of GPM using Omniglot and CIFAR10 images in Fig. 3a. The generation process consists of 2 main steps: (1) sample images to form an episode and write it to the memory; (2) sample $\mathbf{w} \sim p(\mathbf{w})$ ($\mathbf{w}$ is a vector of size $1 \times K$) and take $\mathbf{z} = \mathbf{wM}$ to be the read-out vector. The generated images will be improved via iterative reading. We observe that GPM can generate images that do not appear in the written episode. This phenomenon can be viewed in Fig. 3a-left, where we compare generated patterns with their most similar ones in terms of Hamming distance. Here the generated images with additional details are generally different from their nearest neighbors. More interestingly, in Fig. 3a-right, we can see a generated aircraft image that inherits the background of the frog image (the 3rd row). Theoretically speaking, these new images are formed from spurious local minima, as discussed in Section 2.4.

## 3.4 GPM AS A HUGE STORAGE

We conduct experiments to test the memory capacity of GPM. The experiments still revolve around reconstructing, denoising and generating images, yet on larger episode length. We use the whole Omniglot dataset with $24,345$ training images to train GPM for around $24,000$ iterations for all

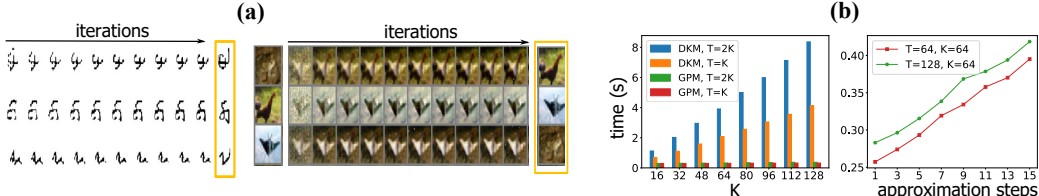

Figure 3: **(a)** *GPM's generated samples.* Images in the box are nearest images in the episode compared with final generated images on their left. Left: *Omniglot patterns.* Distance between images is measured by the Hamming distance. Right: *CIFAR10 images.* Images in the leftmost are the episode written to memory. Each image is repeated 3 times to help the generations better. We use Euclidean metric to measure distance between images. **(b)** *Running time per iteration.* Left: Running time per iteration comparison between GPM and DKM. GPM has a time complexity of almost $O(1)$ with respect to the episode size and memory size. Right: Running time per iteration of GPM with different number of pseudo-inverse approximation steps.

experiments. For convention, we denote the $T$-GPM as the GPM trained on episodes of length $T$. Results are shown in Fig. 4.

In the reconstruction experiment, we keep the memory size fixed and increase the episode length from minimum (equivalent to the number of memory slots) to maximum ($24,345$, which is the whole set of training images). We run the experiment with two memory configurations: $8 \times 100$ and $32 \times 100$. With the same episode length, the $32 \times 100$ memory always performs better, which is reasonable since bigger memory enables better storage capability. We also observe a more interesting phenomenon in both configurations: from a certain point, the reconstruction loss seems to be stable even when we double the episode length. This agrees with earlier discussion in Section 2.2, where we argue that GPM can handle big episodes thanks to the orthogonality of random vectors in high-dimensional space.

We also test the denoising and generating capabilities of GPM when trained with a large episode size. We are curious whether GPM can keep these abilities (to some extent) when storing lots of information. We use the $24,345$-GPM with $32 \times 100$ memory in these experiments. At test time, we first write the whole test set of $8070$ images to the memory and keep this posterior memory fixed, then we either input noisy queries for denoising or sample from the memory for generating images. Despite the huge amount of information compressed into the memory, the model can still denoise and generate images with moderate quality. The average Hamming error when denoising is $97.08$, which is approximately $12.4\%$ of the total pixels.

We note that other baselines such as KM and DKM are inapplicable to such a large-scale scenario due to their slow memory mechanisms. In contrast, GPM can be trained efficiently even with episodes of $24,345$ images. More experiments regarding this point are conducted in the next section.

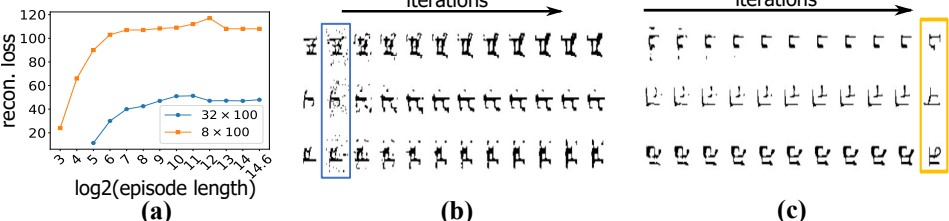

Figure 4: **(a)** Reconstruction loss of GPM with fixed memory size and increasing episode length. **(b)** Denoising salt and pepper noise (5%) of the $24,345$-GPM with $32 \times 100$ memory. Patterns in the first column are original ones; patterns in the blue box are noisy queries; following columns are denoising process during iterative reading. **(c)** Generated patterns of the $24,345$-GPM with $32 \times 100$ memory. Patterns in the orange box are closest patterns written to the memory compared with generated patterns on their left.

## 3.5 ABLATION: RUNNING TIME

GPM is designed to avoid the computation complexity $O(K^3)$ of inverting matrices and $O(T)$ of online Bayesian inference. We conduct experiments to show the time efficiency of GPM compared to DKM in practice. All operations are computed on a single GPU. We use the `inverse` function of Pytorch 1.8.0 (Paszke et al., 2017) for batch matrix inverse. Results are shown in Fig. 3b.

In the first experiment, we run GPM with 7 approximation steps for the pseudo-inverses and compare the running time against DKM in different cases of episode and memory size. As expected, the running time of DKM increases as the memory size $K$ increases while GPM maintains almost the same performance. Moreover, we can see a problem of Bayesian update on the running time: if we double the episode size and keep the same memory size, the running time also doubles. In contrast, it only yields a modest increase in the running time of GPM.

In the second experiment, we would like to see how the number of pseudo-inverse approximation steps affects the running time of GPM in two cases: $T = 64, K = 64$ and $T = 128, K = 64$. The running time curves in the two cases share a similar trend: they depend linearly on the number of approximation steps. Hence, it is important to choose an appropriate initial term for the approximation sequence to reduce the number of approximation steps. More discussion is given in Appendix G.

## 4 RELATED WORK

Memories are known to be critical to tasks that demand long-term dependencies (Hochreiter and Schmidhuber, 1997; Graves et al., 2016; Santoro et al., 2016; Vaswani et al., 2017; Le et al., 2019; Munkhdalai et al., 2019). When all stored data are available to retrieve through attention mechanism (Sukhbaatar et al., 2015; Vaswani et al., 2017), we indeed perform retrieval over a content-addressable memory such as Hopfield networks (Hopfield, 1982; Ramsauer et al., 2021) and Kanerva's sparse distributed memory (SDM) (Kanerva, 1984; Bricken and Pehlevan, 2021).

These works are primarily designed for storing and retrieval of patterns, mostly in a deterministic manner, and thus less concerned about noise and data generation (Le et al., 2018). Kanerva machines (Wu et al., 2018a), inspired by the addressing mechanism of SDM, are generative memory models which fill these gaps. Here memory update/retrieval and addressing mechanism are treated as Bayesian inference where posteriors are updated when a new data episode arrives. This iterative and dynamic inference handles noisy inputs better due to the convergence of the attractor dynamics to fixed points (Wu et al., 2018b). The Kanerva machines thus belong to a class of latent variable models with a very expressive latent prior. These generalize Variational AutoEncoders (VAE) (Kingma and Welling, 2014), deep encoder-decoder neural networks that can generate high-quality data in high dimensional space. VAEs are very powerful when equipped with a high-capacity decoder (Gregor et al., 2015; Bachman, 2016; Ma et al., 2019; Sadeghi et al., 2019) and a expressive prior (Chen et al., 2017; Graves et al., 2018; Tomczak and Welling, 2018). Here memory-based priors in Kanerva machines offer a systematic approach to bring more representation power to VAEs.

However, Bayesian inference in the original Kanerva machines is expensive. To address this drawback, Marblestone et al. (2020) introduced the Product Kanerva Machine to reduce the heavy cost of inverting matrices in the original model. An enhanced variant known as Kanerva++ (Ramapuram et al., 2021) uses a more powerful encoder and regional memory writing-reading to help improve both the retrieval accuracy and training time.

Our work takes a radically alternative view in that we treat memory update as seeking least-square solutions to linear systems. This permits the use of fast matrix pseudo-inverse for the memory operations (Ben-Israel and Greville, 2001). Indeed the pseudo-inverse has been studied in the associative memory literature. Examples include Kohohen-type linear associative memory, which can be referred as a pseudo-inverse neural network (Kohonen, 2012). Personnaz et al. (1986) proposed the projection rule (other than Hebb's learning rule in standard Hopfield Network) involving matrix pseudo-inverses to guarantee perfect storage and retrieval. However, the equilibrium point produced by the projection rule may not be asymptotically stable in the sense that adding new vectors may affect the existing equilibria in the network. To overcome this problem, Yen and Michel (1991) introduced a learning and forgetting algorithm with efficient computation. As far as we are aware, the pseudo-inverse is not commonly used in deep memory networks partly because of the difficulty to compute it exactly, although several attempts have been made to improve the computation efficiency (Greville, 1960; Ben-Israel and Cohen, 1966; Courrieu, 2008; Toutounian and Ataei, 2009).

## 5 DISCUSSION

We have proposed Generative Pseudo-Inverse Memory (GPM) to overcome crucial limitations of previous generative memory models. GPM is not only able to operate fast but also store a large amount of data at a time while maintaining abilities of denoising corrupted patterns and generating

novel patterns. We derived an associated energy function and showed that iterative updates in GPM induce attractor dynamics converging toward fixed points. We demonstrated these capacities on an extensive set of experiments, obtaining superior results.

These new capacities of GPM as a generative memory open up new exciting rooms of investigation. A primary direction is positioning GPM as working memory whose size and processing speed are known to be critical for high-level reasoning (Kyllonen and Christal, 1990), i.e., the capacity to deduce new knowledge through manipulating the old. Here the query encoder will support arbitrary query format, and the decoder will support answer generation. As GPM is designed for episodic handling, streaming data can be periodically cached through local registers before sending to GPM, as suggested in slot-based memories (Le et al., 2019). Finally, GPM may offer fast episodic learning and generative exploration in RL agents.

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

# APPENDIX

## A    RECONSTRUCTIONS

We provide visualizations for the reconstructions of GPM on numerous datasets in Figure 5. We also provide empirical evidence to show that the optimization problem associated with GPM (Eq. (4)) is more direct than DKM (Eq. (3)) in Figure (6): the cosine similarity between stored vectors and read-out's of GPM is nearly 1, which is much better than DKM (around 0.82). We report the ELBO of DKM and GPM over 3 datasets, including FashionMNIST, CIFAR100 and CelebA in Table 3.

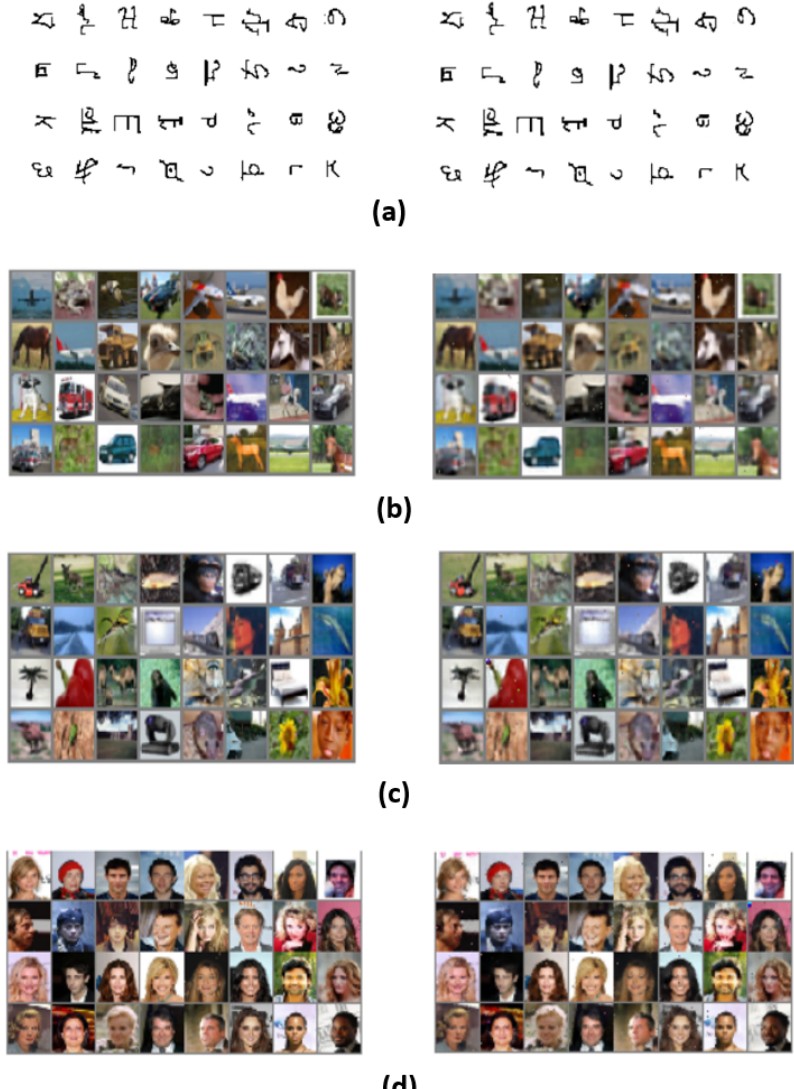

Figure 5:  Reconstructions of GPM. *Left:* original images. *Right:* reconstructions. **(a)** Omniglot **(b)** CIFAR10 **(c)** CIFAR100 **(d)** CelebA

## B    DENOISING

We demonstrate denoising capibility of GPM in Fig. 7. The denoising process consists of two steps: we first write the correct patterns into the memory, then read from the memory using noisy queries. The image will be completed during iterative reading. We also report the full results of the denoising experiment described in Section 3.2 in Figure 8.

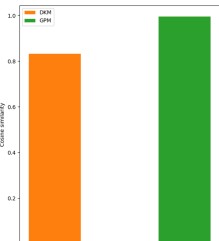

Figure 6: *Average cosine similarity between exact data and read-out's in DKM and GPM*. Train and test on Omniglot dataset.

| Method | FashionMNIST $28 \times 28$ (bits/dim) | CIFAR100 $32 \times 32$ (bits/dim) | CelebA $192 \times 192$ (bits/dim) |
|---|---|---|---|
| DKM (Wu et al., 2018b) | 3.97[†] | 4.88 | 4.71[†] |
| Kanerva++ (Ramapuram et al., 2021) | 3.40 | - | - |
| GPM (ours) | 3.96 | 4.05 | 4.63 |

Table 3: *Negative evidence lower bound of test likelihood (lower is better)* on FashionMNIST, CIFAR100 and CelebA datasets. [†] Our implementation.

## C  TRAINING DETAILS

The encoder consists of 4 layers, each of which is a convolution layer with $4 \times 4$ filter with stride 2 followed by a Resnet block with bottleneck (He et al., 2016). The decoder is simply a mirror of the encoder with transpose convolutional layer. We use the swish activation function (Ramachandran et al., 2017) non-linear layers. For binarized datasets, we use 16 filters for all convnets; for real-valued datasets, we use 256 filters to adapt with the complexity of the data.

The Omniglot is a dataset with great diversity, consists of 1623 different classes and 20 images in each class. Following settings in (Burda et al., 2016), the dataset is splitted into 24,345 training and 8,070 test examples. We use a $32 \times 100$ memory for this dataset. The data noise when writing is sampled from $\mathcal{N}(\mathbf{0}, 0.5^2\mathbf{I})$. At each training step, we randomly sample 32 images to form an episode. We run the Ben-Cohen algorithm for 7 steps to approximate the pseudo-inverses, with the initial term is $10^{-3}$ times the transpose of the matrix which we want to calculate the pseudo-inverse. We use the Bernoulli likelihood for all binarized datasets.

For real-valued datasets, we use a $128 \times 512$ memory with data noise sampled from $\mathcal{N}(\mathbf{0}, 0.5^2\mathbf{I})$. All other settings are the same with the Omniglot dataset, except for the initial terms of the Ben-Cohen algorithm: $10^{-4}$ times the transpose of the matrix when approximating pseudo-inverse of the weight matrix and $5 \cdot 10^{-5}$ times the transpose of the matrix when approximating pseudo-inverse of the memory. We use the discretized mixture of logistics (Salimans et al., 2017) for the output distribution.

In all experiments, we use the Adam optimizer with learning rate varying from $5 \cdot 10^{-5}$ to $5 \cdot 10^{-4}$ depending on the dataset. We use weight decay of $10^{-3}$ along with gradient clipping at threshold 10. For real-valued datasets, we additionally use a scheduler to reduce the learning rate by $0.99995$ after each training epoch. We train all models for 500-1000 epochs with exception 30 epochs for CelebA dataset.

## D  PROPERTIES OF PSEUDO-INVERSE

In this section, we outline the definition and some important properties of the pseudo-inverse of a matrix. Most of the proofs of following propositions and theorems can be found in (Ben-Israel and Greville, 2001).

**Definition D.1.** For every real matrix $\mathbf{A}$ (square or rectangular), there is a unique matrix $\mathbf{Y}$ satisfying all following conditions: $\mathbf{AYA} = \mathbf{A}, \mathbf{YAY} = \mathbf{Y}, (\mathbf{AY})^\top = \mathbf{AY}$ and $(\mathbf{YA})^\top = \mathbf{YA}$, where $^\top$ denotes the transpose of a matrix. The matrix $\mathbf{Y}$ is called the *pseudo-inverse* of $\mathbf{A}$, denoted by $\mathbf{A}^+$.

If $\mathbf{A}$ is a square matrix and non-singular, then $\mathbf{A}^+$ is the inverse $\mathbf{A}^{-1}$ of $\mathbf{A}$. If $\mathbf{A}$ is rectangular of size $m \times n$, then $\mathbf{A}^+$ is of size $n \times m$. The following propositions shows some important relations between $\mathbf{A}$ and $\mathbf{A}^+$.

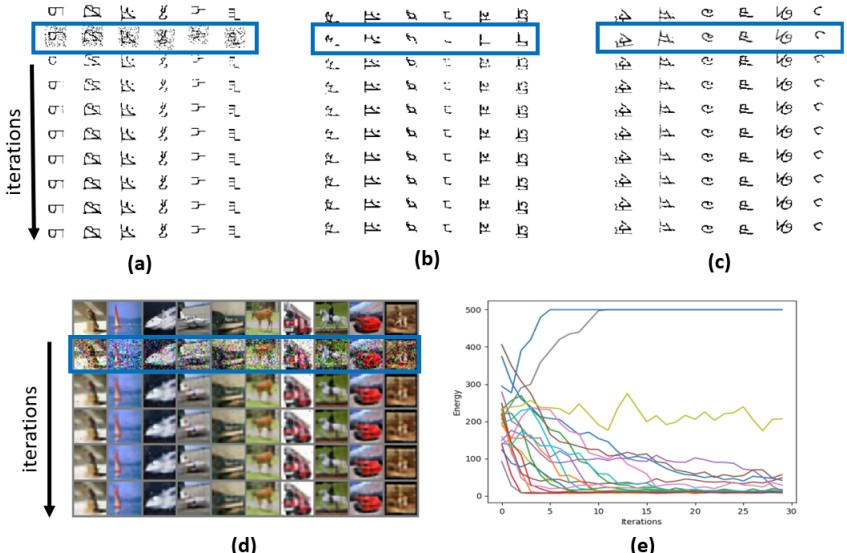

**(a)**                **(b)**                **(c)**

**(d)**                **(e)**

Figure 7: *Retrieving pattern with noisy query during iterative reading*. Top rows are correct patterns; rows in the boxes are corrupted patterns; following rows are retrieval process with iterative reading. **(a)** Salt and pepper noise (15%). **(b)** Block noise ($12 \times 12$ block) to cover a part of the correct pattern. **(c)** Rotation noise (rotation angle uniformly selected from -45 to 45 degrees). **(d)** Gaussian noise with standard deviation 0.5. **(e)** Illustration of energy decrease during retrieving Omniglot patterns. Although sometimes the energy diverges and iterative reading leads to a meaningless pattern, overall, the energy decreases during iterations.

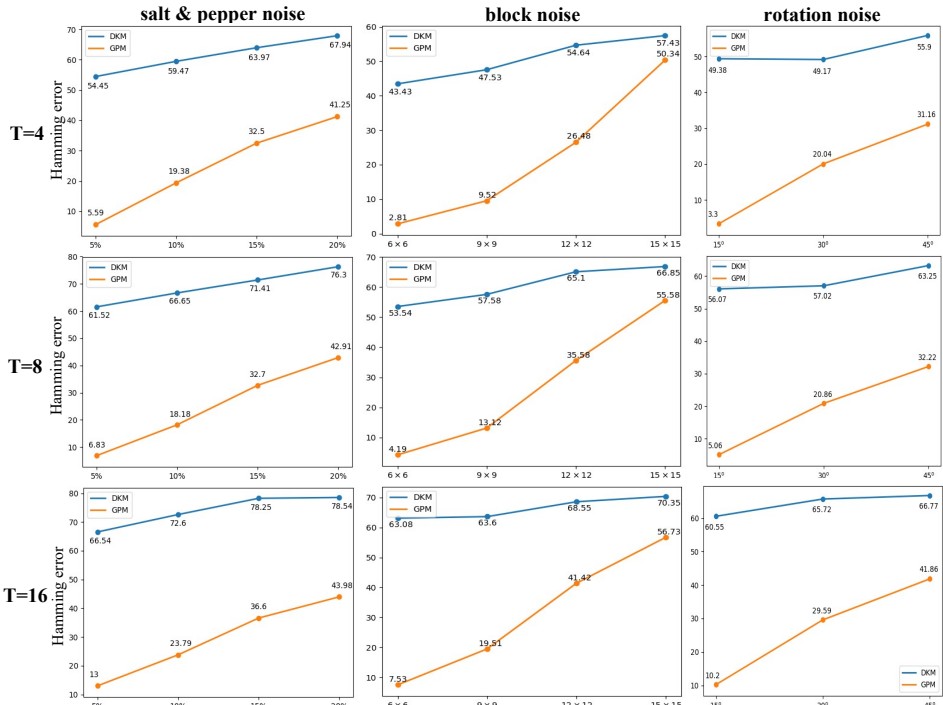

Figure 8: *Hamming error measuring the denoising capability of GPM and DKM (lower is better)*. Experiments are conducted on binarized Omniglot dataset. We test denoising capability of both models with different types of noise (salt & pepper noise, block noise, rotation noise) and levels of noise ($5\% - 20\%$ salt & pepper noise, block noise with block size from $6 \times 6$ to $15 \times 15$, rotation noise with rotation angle from $15°$ to $45°$).

**Proposition D.2.** *Let* $\mathbf{A}$ *be a* $m \times n$ *real matrix. Then*

1. $\mathbf{A}^+\mathbf{A} = \mathbf{I}_n$ if and only if rank$\mathbf{A} = n$ (i.e. columns of $\mathbf{A}$ are linearly independent), where $\mathbf{I}_n$ is the $n \times n$ identity matrix;
2. $\mathbf{A}\mathbf{A}^+ = \mathbf{I}_m$ if and only if rank$\mathbf{A} = m$ (i.e. rows of $\mathbf{A}$ are linearly independent), where $\mathbf{I}_m$ is the $m \times m$ identity matrix.

**Proposition D.3.** *For every real matrix* $\mathbf{A}$*:*

1. $(\mathbf{A}^+)^+ = \mathbf{A}$;
2. $\left(\mathbf{A}^\top\right)^+ = \left(\mathbf{A}^+\right)^\top$;
3. $\mathbf{A}^+ = \left(\mathbf{A}^\top\mathbf{A}\right)^+ \mathbf{A}^\top = \mathbf{A}^\top\left(\mathbf{A}\mathbf{A}^\top\right)^+$.

If $\mathbf{A}$ and $\mathbf{B}$ are square invertible matrices such that $\mathbf{A}\mathbf{B}$ is also invertible, then we have the well-known property $(\mathbf{A}\mathbf{B})^{-1} = \mathbf{B}^{-1}\mathbf{A}^{-1}$. The same property does not always hold for pseudo-inverse. However, with some sufficient condition of independence, we will get the desired property.

**Proposition D.4.** *If* $\mathbf{A}$ *has linearly independent columns or* $\mathbf{B}$ *has linearly independent rows, then* $(\mathbf{A}\mathbf{B})^+ = \mathbf{B}^+\mathbf{A}^+$.

**Theorem D.5.** *For any real matrix* $\mathbf{A}$ *of size* $m \times n$*, the limit* $\lim_{\sigma \searrow 0} \mathbf{A}^\top(\mathbf{A}\mathbf{A}^\top + \sigma\mathbf{I}_m)^{-1}$ *always exists. Moreover,* $\lim_{\sigma \searrow 0} \mathbf{A}^\top(\mathbf{A}\mathbf{A}^\top + \sigma\mathbf{I}_m)^{-1} = \mathbf{A}^+$.

The concept of pseudo-inverse arose when ones tried to solve the linear system $\mathbf{A}\mathbf{x} = \mathbf{b}$, where $\mathbf{b}$ is some vector with appropriate size. When the system has no solution, it is reasonable to find some vector $\mathbf{x}$ that minimizes the Euclidean norm of the residual $\mathbf{r} = \mathbf{b} - \mathbf{A}\mathbf{x}$. Any solution of the optimization problem $\min_{\mathbf{x}} \|\mathbf{b} - \mathbf{A}\mathbf{x}\|_F$ (where $\|.\|_F$ is the Frobenius norm) is called the *least-square solution* of the system $\mathbf{A}\mathbf{x} = \mathbf{b}$. It can be proven that $\mathbf{x} = \mathbf{A}^+\mathbf{b}$ is a least-square solution. Moreover, among the least-square solutions of system $\mathbf{A}\mathbf{x} = \mathbf{b}$, $\mathbf{x} = \mathbf{A}^+\mathbf{b}$ is the one of minimum Euclidean norm, i.e. the *minimum-norm least-square solution*. This result can be generalized as in the following proposition.

**Proposition D.6.** *Let* $\mathbf{A}, \mathbf{B}, \mathbf{D}$ *be real matrices with appropriate size. The minimum-norm least-square solution of the system* $\mathbf{A}\mathbf{X}\mathbf{B} = \mathbf{D}$ *is* $\mathbf{X} = \mathbf{A}^+\mathbf{D}\mathbf{B}^+$.

## E   PROOFS OF THEORETICAL RESULTS ON ERROR BOUNDS

We provide the proofs for Theorem 2.2 and Corollary 2.3.

*Proof of Theorem 2.2.* For convention, let us denote $\mathbf{E}_{\mathbf{W}_0} = \mathbf{W}_0\mathbf{W}_0^+ - \mathbf{I}$. Following Algorithm 1, we get

$$\begin{aligned}
\overline{\mathbf{W}}\mathbf{M} &= \mathbf{Z}\mathbf{Z}_\xi^+\mathbf{W}_0\mathbf{W}_0^+\mathbf{Z}_\xi \\
&= \mathbf{Z}\mathbf{Z}_\xi^+(\mathbf{I} + \mathbf{E}_{\mathbf{W}_0})\mathbf{Z}_\xi \\
&= \mathbf{Z}\left[\mathbf{I} + (\mathbf{Z}_\xi^+\mathbf{Z}_\xi - \mathbf{I}) + \mathbf{Z}_\xi^+\mathbf{E}_{\mathbf{W}_0}\mathbf{Z}_\xi\right].
\end{aligned}$$

Since $\|\mathbf{I} - \mathbf{Z}_\xi\mathbf{Z}_\xi^\top\|_2 \leq \alpha$, by (Rump, 2011, Lemma 2.2), it follows $\|\mathbf{Z}_\xi\|_2 \leq \sqrt{1+\alpha}$ and $\|\mathbf{Z}_\xi^+\|_2 \leq \frac{1}{\sqrt{1-\alpha}}$. Since $\|.\|_2$ is a multiplicative norm, we finally have

$$\left\|(\mathbf{Z}_\xi^+\mathbf{Z}_\xi - \mathbf{I}) + \mathbf{Z}_\xi^+\mathbf{E}_{\mathbf{W}_0}\mathbf{Z}_\xi\right\|_2 \leq \left\|\mathbf{Z}_\xi^+\mathbf{Z}_\xi - \mathbf{I}\right\|_2 + \left\|\mathbf{Z}_\xi^+\mathbf{E}_{\mathbf{W}_0}\mathbf{W}_\xi\right\|_2$$

$$\leq \beta + \|\mathbf{E}_{\mathbf{W}_0}\|_2\sqrt{\frac{1+\alpha}{1-\alpha}},$$

which is our desired result.

*Proof of Corollary 2.3*: This is a special case of Theorem 2.2. When rows of $\mathbf{W}_0$ and $\mathbf{Z}$ are independent, $\mathbf{W}_0\mathbf{W}_0^+ = \mathbf{I}$ and $\mathbf{Z}_\xi\mathbf{Z}_\xi^+ = \mathbf{I}$, which leads to $\mathbf{E}_{\mathbf{W}_0} = 0$ and $\beta = 0$.

## F   ITERATIVE READING FROM THE VIEW OF FIXED POINT

In addition to the discussion in Section 2.4, we can alternatively think about the attractor as the fixed point of the reading operation. Assume that the current query is $\mathbf{q}_t$ and the next query is determined

by $\mathbf{q}_{t+1} = f(\mathbf{q}_t)$, where $f$ represents for the reading operation. If $\{\mathbf{q}_t\}$ converges to some $\mathbf{q}^*$, then $\mathbf{q}^* = f(\mathbf{q}^*)$, which means $\mathbf{q}^*$ is a fixed point of $f$. If the model is well-trained, stored patterns are fixed points of $f$. (Banach, 1922) provides a sufficient condition for an iterative sequence to converge to a fixed point. Roughly speaking, if $\mathbf{q}^*$ is a fixed point of $f$ such that $\|\mathbf{J}(\mathbf{q}^*)\|_2 < 1$ (where $\mathbf{J}$ is the Jacobian of $f$) and the whole sequence $\{\mathbf{q}_t\}_{t\geq 0}$ locates in a small neighborhood of $\mathbf{q}^*$, then $\mathbf{q}_t \to \mathbf{q}^*$. This result demonstrates the ability to converge to a correct pattern given a good enough initial query. Similar to spurious local minima in the energy landscape, there also exists many spurious fixed points in the $f$ landscape (see Fig. 9). This phenomenon is not expected when retrieving, but it may be helpful in case we want to generate new images as demonstrated in Section 3.3.

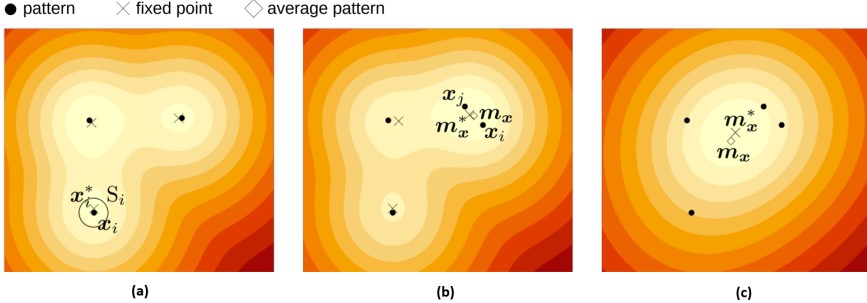

Figure 9: *The three cases of fixed points.* **(a) Fixed point is single pattern:** patterns are stored if they are well separated. Each pattern $x_i$ has a single fixed point $x_i^*$ close to it. **(b) Fixed point is average of similar patterns:** $x_i$ and $x_j$ are similar to each other and not well separated. The fixed point $m_x^*$ is a spurious fixed point that is close to the mean $m_x$ of the similar patterns. **(c) Fixed point is average of all patterns:** no pattern is well separated from the others. A single global fixed point $m_x^*$ exists that is close to the arithmetic mean $m_x$ of all patterns. Images and caption are taken from (Ramsauer et al., 2020).

## G    Initial condition for Ben-Cohen algorithm

Suppose $\mathbf{A}$ is a real matrix. With $0 < \alpha < \frac{2}{\sigma_1^2}$, where $\sigma_1$ is the largest eigenvalue of $\mathbf{A}\mathbf{A}^\top$, the sequence defined as $\mathbf{A}_0 = \alpha\mathbf{A}^\top$, $\mathbf{A}_{i+1} = 2\mathbf{A}_i - \mathbf{A}_i\mathbf{A}\mathbf{A}_i$ will converge to $\mathbf{A}^+$ (Ben-Israel and Cohen, 1966). This choice of initial term may lead to slow convergence when the condition number $\kappa(\mathbf{A}) = \|\mathbf{A}\|_2\|\mathbf{A}^+\|_2$ is large (Söderström and Stewart, 1974). If we start with an $\mathbf{A}_0$ such that $\mathbf{A}_0\mathbf{A} = (\mathbf{A}_0\mathbf{A})^\top$ and $\mathbf{A}_0$ is already close to $\mathbf{A}^+$, for example $\mathbf{A}_0 = \mathbf{A}^\top(\mathbf{A}\mathbf{A}^\top + \delta\mathbf{I})^{-1}$ where $\delta > 0$ and small enough, then $\{\mathbf{A}_i\}_{i\geq 0}$ converges quadratically to $\mathbf{A}^+$. However, this requires us to calculate the inverse of $\mathbf{A}\mathbf{A}^\top + \delta\mathbf{I}$, which we tend to completely avoid in GPM. Therefore we keep the original setting of Ben-Cohen for the initial term and let the model adapt with the approximation error.

## H    Solution of Bayesian update optimization problem

Let $\mathbf{M}_0, \mathbf{M}$ denote random variables of the prior and posterior memory, respectively, and suppose that both have matrix Gaussian distributions. Specifically, $\mathbf{M}_0 \sim \mathcal{N}(\mathbf{R}_0, \mathbf{U}_0, \mathbf{I})$ and $\mathbf{M} \sim \mathcal{N}(\mathbf{R}, \mathbf{U}, \mathbf{I})$. Given the prior $\mathbf{M}_0$, the to-be-written data $\mathbf{Z}$ and the writing weight $\mathbf{W}_0$, the optimization problem

$$\min_{\mathbf{A},\mathbf{B}} \mathbb{E}\left(\|\mathbf{M} - \mathbf{M}_0\|_F^2\right) \quad \text{s.t.} \quad \mathbf{M} = \mathbf{A}\mathbf{Z} + \mathbf{B} \text{ and } \mathbf{Z} = \mathbf{W}_0\mathbf{M}_0 + \xi$$

has a unique solution $\mathbf{M} \sim \mathcal{N}(\mathbf{R}, \mathbf{U}, \mathbf{I})$, where

$$\mathbf{R} = \mathbf{R}_0 + \mathbf{U}_0^\top\mathbf{W}_0^\top\left(\mathbf{W}_0\mathbf{U}_0\mathbf{W}_0^\top + \Sigma_\xi\right)^{-1}(\mathbf{Z} - \mathbf{W}\mathbf{R})$$

$$\text{and} \quad \mathbf{U} = \mathbf{U}_0 - \mathbf{U}_0^\top\mathbf{W}_0^\top\left(\mathbf{W}_0\mathbf{U}_0\mathbf{W}_0^\top + \Sigma_\xi\right)^{-1}\mathbf{W}_0\mathbf{U}_0,$$

where $\Sigma_\xi$ is diagonal matrix whose diagonal elements are $\sigma_\xi^2$.

## I    Derivation for evidence lower bound

The evidence lower bound of the conditional log-likelihood $p(\mathbf{X}|\mathbf{M})$ can be derived as

$$\ln p(\mathbf{X}|\mathbf{M}) = \mathbb{E}_{q(\mathbf{W})}\left[\ln\frac{p(\mathbf{X}|\mathbf{M})q(\mathbf{W})}{q(\mathbf{W})}\right]$$

$$= \mathbb{E}_{q(\mathbf{W})}\left[\ln\frac{p(\mathbf{X}|\mathbf{W},\mathbf{M})p(\mathbf{W}|\mathbf{M})q(\mathbf{W})}{p(\mathbf{W}|\mathbf{X},\mathbf{M})q(\mathbf{W})}\right]$$

$$= \mathcal{L} + D_{\mathrm{KL}}(q(\mathbf{W})||p(\mathbf{W}|\mathbf{X},\mathbf{M})).$$

## J    RELATION WITH EM ALGORITHM

We show that GPM can be viewed as 2-step EM algorithm (Dempster et al., 1977). Let us remind ourselves the log-likelihood approximation derived in Section 2.1:

$$\ln p(\mathbf{X}|\mathbf{M}) = \mathbb{E}_{q(\mathbf{W})}\ln p(\mathbf{X}|\mathbf{W},\mathbf{M}) - D_{\mathrm{KL}}(q(\mathbf{W})||p(\mathbf{W})) + D_{\mathrm{KL}}(q(\mathbf{W})||p(\mathbf{W}|\mathbf{X},\mathbf{M})). \quad (8)$$

Keeping $p(\mathrm{vec}(\mathbf{W})) \sim \mathcal{N}(\mathbf{0}, \mathbf{I})$ in mind, we can rewrite the conditional log-likelihood as

$$\ln p(\mathbf{X}|\mathbf{M}) = \mathbb{E}_{q(\mathbf{W})}\ln p(\mathbf{X}|\mathbf{W},\mathbf{M}) + \frac{\|\mathbf{Z} - \overline{\mathbf{W}}\mathbf{M}\|_F^2}{2\sigma_\xi^2} + \mathbf{R}, \quad (9)$$

where $\mathbf{R}$ is residual term that is independent of $\overline{\mathbf{W}}$. From the perspective of EM algorithm and Eq. ((9)), Algorithm 1 can be thought as a 2-step process to maximize the log-likelihood:

1. Keeping the likelihood unchanged and tightening the bound by minimizing $\|\mathbf{Z} - \overline{\mathbf{W}}\mathbf{M}\|_F^2$;
2. Maximizing $\mathbb{E}_{q(\mathbf{W})}\ln p(\mathbf{X}|\mathbf{W},\mathbf{M})$, i.e. the negative reconstruction loss.

Note that in training, we maximize the evidence lower bound $\mathcal{L}_T = \mathbb{E}_{q(\mathbf{W})}\ln p(\mathbf{X}|\mathbf{W},\mathbf{M}) - D_{\mathrm{KL}}(q(\mathbf{W})||p(\mathbf{W}))$ instead of $\mathbb{E}_{q(\mathbf{W})}\ln p(\mathbf{X}|\mathbf{W},\mathbf{M})$. Therefore, for the maximum values of $\mathcal{L}_T$ and $\mathbb{E}_{q(\mathbf{W})}\ln p(\mathbf{X}|\mathbf{W},\mathbf{M})$ to be as close as possible, we look for $\overline{\mathbf{W}}$ that minimizes $D_{\mathrm{KL}}(q(\mathbf{W})||p(\mathbf{W}))$. Since $p(\mathrm{vec}(\mathbf{W})) \sim \mathcal{N}(\mathbf{0}, \mathbf{I})$, this is equivalent to seeking $\overline{\mathbf{W}}$ with minimum norm. This fits with the step of finding the weight matrix $\overline{\mathbf{W}}$ in the reading phase, where we determine $\overline{\mathbf{W}} = \mathbf{Z}\mathbf{M}^+$, which is the least-square minimum-norm solution of the linear system $\overline{\mathbf{W}}\mathbf{M} = \mathbf{Z}$.

