# OpenReview forum: "Generative Pseudo-Inverse Memory"
_ICLR.cc/2022/Conference — ICLR 2022 Poster_

### Official Review · Reviewer_DcwF · 2021-11-03

**Correctness:** 3
**Technical Novelty And Significance:** 2
**Empirical Novelty And Significance:** 2
**Recommendation:** 5
**Confidence:** 2

**Main Review:**

## Strengths
   - The proposed method seems to perform much better than competing methods on the toy experiments

## Weaknesses
   - In my opinion, the authors don't provide a clear enough review of the relevant literature. The only section with some detail on related work is section 2.2.1, and connections are only drawn to DKMs and the EM algorithm. Given that GPM is compared against several methods in Table 1, none of which are described in any depth besides KMs and DKMs it would be useful to provide some context for these methods.
   - Some of the comparisons in Table 1 seem a little strange. For example, every method in the "Improved decoders" section outperforms GPM in terms of negative ELBO on CIFAR-10, which is arguably a more relevant (though still toy) problem. Based on this, I would expect that these methods would outperform GPM on CIFAR-100 and CelebA too, however, these numbers were omitted from the table and instead GPM was proclaimed as the best. I think if the authors want to compare against these methods, then either they should include results for the other datasets as well, omit the comparison or not conclude GPM as the best on datasets on which performance metrics for the other methods are unavailable.
   - It is unclear to me what is being reported as negative ELBO in Table 1 for the memory models. Is it the quantity from Eq (1) or Eq (2)?
   - I find the results for VAEs, IWAEs and the "Richer priors" section in Table 1 superfluous, can the authors clarify why these were included?
   - My current understanding is that the "only" difference from DKMs is that GPM optimizes the objective in Eq (4) instead of Eq (3) and uses fast approximations to the pseudo-inverses of matrices involved in the memory update step. Given this, it is unclear to me how the large improvements reported in Table 1 from DKM to GPM can arise. Could the authors comment on this?
   - Apart from Figure 1, every figure needs improvement. Especially in Figure 4, without zooming heavily the axes are completely unreadable. In my opinion, Figure 3 and Figure 4b and 4c are trying to show too much and the result is cluttered. I think it would be more valuable to include fewer rows with larger images.

### Minor items
   - in the paragraph below Eq 1 the sentence contains a mistake, the KL in Eq is from $q(W)$ to $p(W)$ and not from $p(W)$ to $q(W)$
   - I believe that on a few occasions the authors use the word "quantize" to mean "quantify"


**Summary Of The Paper:**

The authors propose a novel deep generative memory model called Generative Pseudo-inverse Memory (GPM) that extends Dynamic Kanerva Machines (DKM, [1]). These models are both deep generative models that maintain a hidden state similar to recurrent neural networks called memory and have ways of writing into and reading from this memory. A key limitation of DKMs is that the time complexity of updating the memory scales cubically in the memory dimensionality and linearly in the episode length. The authors observe that by slightly altering the optimization objective that GPM has to solve, under mild assumptions the memory update can be performed much more efficiently, allowing them to write entire datasets into the memory of the GPM with little computational cost.

The authors perform some experiments on some toy datasets to validate their method and obtain good results against related methods.

[1] Y. Wu, G. Wayne, K. Gregor, and T. Lillicrap. Learning attractor dynamics for
generative memory. NeurIPS 2018

**Summary Of The Review:**

I believe that the theoretical contributions of the paper with respect to DKMs is good if perhaps a bit incremental, however, I found the empirical results questionable.
If the authors can address my questions I will consider raising my score.

---

> ### Author Response · Authors · 2021-11-20
> **Response for Reviewer DcwF**
>
> Thank you for your detailed review and valuable comments. We have revised our manuscript to address your questions and concerns as follows.
>
> • Thank you for suggesting more detailed review of the relevant literature. The space constraint had prevented us from explaining every single related work in detail. In the new revision, we have managed to add the “Related work” section in which we give an overview of memory models, generative models, usages of the pseudo-inverses in neural networks, and contrasting our work against existing works.
>
> • On comparisons in Table 1: Through the experiments, our main goal is to show that appropriately incorporating an external memory with pseudo-inverse mechanism can help boost both the accuracy and speed of the model, compared to previous memory models such as the Kanerva Machines. To have a fair comparison, we only incorporate standard pairs of encoder-decoder (to be specific, concatenations of Resnet blocks only) into the GPM, and even in such cases, GPM outperforms the Kanerva Machines with large margins. We believe that if GPM is equipped with stronger decoders or richer priors, it will outperform memoryless models with similar decoders or priors.
>
> • The negative ELBO reported in Table 1 is the quantity from Eq. (1). Eq. (2) is only used for training. We added a clarification in the revision.
>
> • The VAE and IWAE are included to show the superiority of memory models (GPM, Kanerva Machines, etc.) compared to memoryless ones. The “richer priors” and “improved decoders” sections are included to show that with simple datasets (where the roles of the encoder and decoder are less important), incorporating external memories can help boost the models' performance without using complex techniques. The “memory models” section is included to show the superiority of GPM compared to previous memory models.
>
> • We would like to clarify our contribution with respect to DKMs as follows. Our proposed optimization problem (Eq. (4)) provides a radical alternative to the traditional Bayesian inference approach, as a result offering new capacities not seen before, and these account for the large improvements compared to DKMs. More specifically, we treat the memory writing mechanism in a more direct way: We will write data into the memory in a way such that the read-out's will be similar to the written data (see Eq. (4) and Eq. (5)). It may seem trivial, but the Bayesian inference approach does not ensure such property in practice: Figure 6 in Appendix A shows that read-out's from Bayesian-based memory may not be similar to written data, while read-out's from GPM's memory are almost perfect retrievals. Hence, our approach is not a mere modification of technique; it is instead a different approach to the problem of writing data into the memory.
>
> • We thank you for pointing out the readability of figures. We have tried our best to improve the readability in the revision: We use bigger font sizes for the axes labels, thus making it easier to read without heavily zooming; we also reduce the number of rows in Figure 3.
>
> • For typos and grammars, thank you for your detailed reading. We have corrected the order of q(W) and p(W) in the text below Eq. (1). All “quantize” words are modified to “quantify”.
>
> We hope that our responses will satisfy your concerns about our paper. We believe that your comments have helped us to improve the writing quality substantially.

---

### Official Review · Reviewer_m6LR · 2021-11-06

**Correctness:** 4
**Technical Novelty And Significance:** 3
**Empirical Novelty And Significance:** 3
**Recommendation:** 8
**Confidence:** 4

**Main Review:**

Strength
- The general memory model (e.g., for episodic or working memory) is an important topic in achieving human-like general AI.
- The paper is well-written and adequately clear (but can be improved).
- The experiment is performed thoroughly.
- It shows clear advantages to the previous works.
- It also provides theoretic analysis on error bound.

Weakness
- The explanation can be improved for the readers who are not familiar with the previous works of KMs. For example, each line of Algo 1. can more explicitly explained.
- The meanings of "temporary read-out" and "dynamic" weight were not clear in the beginning.
- In introduction, it says "Importantly, our model ... not only store and retrieve ... but also generate ..." This sounds like a feature of only the proposed model, but the previous works can also do this.

Minor
- It may be beyond the scope of the paper, but showing the actual benefit of this memory in an RL agent will make the paper more complete.

**Summary Of The Paper:**

The paper proposes a new memory model following the research line of Kanerva Machine, DKM, and Kanerva++. The proposed model reformulates the Bayesian updates of memory and address as finding least-square solution to linear systems. This requires matrix inversion operations and the authors proposed to approximate it iteratively by using Ben-Cohen algorithm for pseudo-inverse matrix. It results in a memory read/write system that is rapid/accurate and can store large batch of data. The evaluation is performed thoroughly on various datasets including binarized MNIST, binarized Omniglot, Fashion MNIST, CIFAR10/100, CelebA. It shows the superiority of the proposed method in negative ELBO of test likelihood, denoising success rate and hamming error, generation, memory capacity, and run-time per iteration.


**Summary Of The Review:**

The paper tackles an important problem and is well and clearly written (but clarity can be improved). The proposed idea is valid and interesting. The experiment clearly shows the superiority of the proposed method in rapid memory read/wright, memory capacity, and de-noising, and generation, etc. The proposed method is also supported by a theoretic analysis on error bound.

---

> ### Author Response · Authors · 2021-11-20
> **Response for Reviewer m6LR**
>
> Thank you for your understanding and valuable comments. We have revised our manuscript to address your questions and concerns as follows.
>
> • Thank you for your suggestion on presenting more explanation of previous works of KMs. In the revision, we have added comments to lines of Algorithm 1 to make it clearer for the readers. More importantly, we have added a new “Related work” section (Section 4), where we included an overview of memory models, generative models and usages of the pseudo-inverses in neural networks to help readers better understand the contexts and contributions of our paper.
>
> • In the revision, we have omitted the term “temporary read-out” since it is rarely used in the main text. The term “dynamic weights” means that the (writing and reading) weights of an input (e.g. an image) not only depend on its own but also the episode it belongs to. The same explanation can be found in the footnote when we introduce the writing and reading weights (at the beginning of Section 2).
>
> • In the introduction, we have modified the sentence to avoid misunderstanding that ours is the first to do “not only store and retrieve ... but also generate ..”.
>
> • One your question of evaluating on RL agents, it is an excellent suggestion. We thank you for that. It would be an interesting direction for our future works, and we have added this point into the Discussion section.
>
> We hope that our responses have addressed your concerns on our paper. Thank you very much for your generosity.

---

### Official Review · Reviewer_NH7Y · 2021-11-07

**Correctness:** 3
**Technical Novelty And Significance:** 3
**Empirical Novelty And Significance:** 2
**Recommendation:** 5
**Confidence:** 4

**Main Review:**

(i) The overall conceptualization presented here is neat. The authors propose an efficient implementation for their algorithm, demonstrate performance on a variety of tasks both quantitatively and qualitatively.

(ii) The text, however, is a bit hard to follow at times and needs to be severely improved both clarity-wise and grammatically.
(a) I found the various usages of "prior memories" and "posterior memories" confusing. To me, it seemed to imply that probability densities were explicitly defined on the memories (i.e. Ms). But it turns out that the parameterization was actually on the read in/out weights (Ws) which implicitly defined distributions on the memories. Furthermore in the section "Memory inference" the authors switch to considering deterministic memories. In general, it would be much appreciated if the authors clarified if/when they are estimating true posterior densities.

(b) There are also terminologies presented in the main body of the paper (such as "temporary readout") that are barely re-used throughout the article. It might be better to avoid these (at least while presenting the core ideas) to improve readability.

(c) In Equation 3, Z is presented as a linear function of M_0, but Z is in fact a function of X from Figure 1. Could the authors clarify this mismatch?

(d) The paper can benefit from a thorough grammatical check and proofreading. Examples include:
Pg 1 "This dynamics enables GPM..." --> These dynamics enable GPM...
Pg 2 "dependent of the episode" --> dependent on the episode
Pg 5 "Detail is given in Appendix J" --> Details are given in Appendix J

(iii) This study also warrants proper comparisons to modern variational approaches. The empirical evaluations presented in the study only reports metrics from classic approaches. Is there a reason why Table 1 is incomplete? Also, the functional benefits of the proposed approach versus variational methods aren't detailed explicitly. On a more general note, a dedicated literature review will greatly help.

(iv) One of the main claims of the paper is the reformulation of the Bayesian update as finding the least-squares solution to a linear system. This equivalence isn't explicitly proven though. It would be a worthy addition to the main text.

(v) Could the authors include any quantification of "fixed-point" behavior?

**Summary Of The Paper:**

The authors propose Generative Pseudo-Inverse Memory (GPM), a family of generative models that offer read and write operations of constant time complexity. Encoding new memories and decoding data from memories are postulated as Bayesian updates for which an equivalent minimization problem is proposed. This minimization problem essentially amounts to solving a linear system of equations, which can be efficiently done via computing matrix pseudo inverses. The authors demonstrate the utility of GPM on a variety of applications such as image denoising, image generation, and storage retrieval.

**Summary Of The Review:**

This is a neat and well-thought-out idea. However, as I've expressed in the main review, some of the claims need further justification and the paper clarity needs to be improved. I am willing to update my score if the authors are able to provide a convincing response!

---

> ### Author Response · Authors · 2021-11-20
> **Response for Reviewer NH7Y**
>
> Thank you for your detailed review and valuable comments, especially holding a positive view of our work. We have revised our manuscript to address your questions and concerns as follows.
>
> (ii)
>
> (a) We have added a remark to clarify the terms: “In this paper, we reuse classical terms 'prior memory' and 'posterior memory', which are frequently used in prior works where the memory is modeled as a distribution. In this paper, however, we only consider deterministic memories with an exception when we introduce the optimization problem corresponding to the Bayesian inference (Eq. (3)). Hence, 'prior memory' and 'posterior memory' should be interpreted as the memory before and after data arrive, respectively”. We also notify the readers whenever the probabilistic schema is applied, i.e. at the beginning of the Intuition part of Section 2.2, and when we switch back to the deterministic schema, i.e. at the beginning of the Memory inference part in Section 2.2.
>
> (b), (d) We thank you for pointing out grammatical errors and readability issues. We have fixed possible grammatical errors and avoided redundant terminologies.
>
> (c) There is no mismatch. In the revision, we clarify that the linear constraint over Z is an assumption. This assumption is quite common, e.g. see Eq. (6) in [4] and Eq. (1) in [5]. It helps the Bayesian inference in Kanerva Machines achieve an explicit posterior distribution that can be easily computed. It also helps reduce the optimization problem (4) to (5) (in our paper), which can be solved using the pseudo-inverses.
>
> (iii) To make our study comparable with recent works on generative memory, we only report results for VAE baselines included in the standard benchmark introduced by [6]. We note that Table 1 already contains state-of-the-art generative memory approaches, which is the main focus of our paper. Having said that, we totally agree that a dedicated literature review would be beneficial and we have added a new section on “Related Work” in the revision (Section 4), in that we clarify the advantage of our method over variational approaches. In particular, our Generative Pseudo-inverse Memory does not suffer from slow writing and reading (which is a common bottleneck of not only the Kanerva Machines but also many other Memory-Augmented Neural Networks). This new capability ensures highly accurate retrieval (both empirically and theoretically) and allow us to have a memory capacity up to a dataset. To the best of our knowledge, this is not examined by any previous deep memory models due to costly writing-reading operations.
>
> (iv) We thank you for pointing this out. We have added a reference that presents a result of the relationship between Bayesian inference (or Bayesian update) and the minimum mean square problem [1, page 172]. It suggests that we can use tools in optimization to tackle drawbacks of Bayesian models. Our model is an example that utilizes the concept of pseudo-inverse (which is quite well-known in the field of optimization and linear systems) to improve the accuracy and speed of previous Bayesian-based Kanerva Machines.
>
> (v) There are several works analyzing the rates of convergence of fixed-point iterations (for example, see [2, 3]). We also provide a slight analysis of the fixed-point behavior in Appendix F, however, in general it is difficult to provide detailed quantitative analysis since the update function is very complex. We believe this is an important direction for future research, since understanding the behavior of fixed points provides hints to better model design, for example, a model that can widen the basins of attraction around the fixed points which leads to more stable convergence to stored data.
>
> We hope that the above responses are sufficient to present a clearer view of our paper.
>
> REFERENCES
>
> [1] Edwin T Jaynes. Probability theory: The logic of science. Cambridge university press, 2003.
>
> [2] Fathollahi, Shahin et al. “A comparative study on the convergence rate of some iteration methods involving contractive mappings.” Fixed Point Theory and Applications 2015 (2015): 1-24.
>
> [3] Phuengrattana, Withun and Suthep Suantai. “Comparison of the Rate of Convergence of Various Iterative Methods for the Class of Weak Contractions in Banach Spaces.” Thai Journal of Mathematics 11 (2012).
>
> [4] Yan Wu, Greg Wayne, A. Graves, and T. Lillicrap. The Kanerva machine: A generative distributed memory. ICLR, 2018.
>
> [5] Yan Wu, Greg Wayne, Karol Gregor, and Timothy Lillicrap. Learning attractor dynamics for generative memory. NeurIPS, 2018.
>
> [6] Jason Ramapuram, Yan Wu, and Alexandros Kalousis. Kanerva++: extending the kanerva machine with differentiable, locally block allocated latent memory. ICLR, 2021.

---

### Author Response · Authors · 2021-11-30
**Summary of the revision**

We thank the reviewers for their careful and detailed reviews. In this comment, we would like to highlight what has been improved in the revision.

• We have added the “Related work” section in which we give an overview of memory models, generative models, usages of the pseudo-inverses in neural networks, and contrasting our work against existing works.

• We have improved readability of Section 2 by adding a reference for the optimization problem associated with Bayesian inference (Eq. (3)) and introducing the linear assumptions over Z separately from the optimization problems.

• We have clarified or avoided confusing terminologies (i.e. “prior memory”, “posterior memory”, “temporary read-out”). We have also fixed possible grammatical errors.

• We have tried our best to improve the readability in the revision: We use bigger font sizes for the axes labels, thus making it easier to read without heavily zooming; we also reduce the number of images in a figure to make them easier to observe.

Please let us know if there are additional questions/concerns before the end of the discussion period. We would be more than happy to address your comments.

---

### Decision · Program_Chairs · 2022-01-20

**Decision:**

Accept (Poster)

**Comment:**

The authors present a new memory-augmented neural network that is related to the Kanerva machine of Wu et. al.  The reviewers considered the ideas in the paper novel and interesting, but were concerned about presentation issues and literature review.  The authors have improved both... however- authors: please even under limited space constraints, make more room for related work!  Clarifying your contribution in the context of the literature is critical for reader understanding, and neglecting this almost had your paper rejected out of hand.

I am voting to accept